# Survey research of patient's preference on choosing microscopic or endoscopic spine surgery for lumbar discectomy

**Gun Keorochana[1], Chaiwat Kraiwattanapong [1]\*, Thamrong Lertudomphonwanit[1], Umaporn Udomsubpayakul[2], Pittavat Leelapattana[1], Pongsthorn Chanplakorn[1], Nutthee Wannaratsiri[1], Tulyapruek Tawonsawatruk[1]**

**1** Department of Orthopaedics, Ramathibodi Hospital, Mahidol University, Bangkok, Thailand, **2** Section for Clinical Epidemiology and Biostatistics, Ramathibodi Hospital, Mahidol university, Bangkok, Thailand

\* Chaiwatkrai@gmail.com

## Abstract

### Background

There are several surgical methods of lumbar discectomy which provide the similar clinical outcomes. There is no clear evidence for how to select the procedures. To better understand the patient's opinion and decision process in the selection of surgical methods between microscopic lumbar discectomy (MLD) and endoscopic lumbar discectomy (ELD).

### Methods

A cross-sectional survey study. Summary information sheet was created by reviewing the comparative literatures, and tested for quality and bias. Participants read the summary information sheet then were asked to complete the anonymous questionnaire.

### Results

Seventy-six patients (71%) of patients who had no experience in lumbar discectomy selected ELD while 31 patients (29%) selected MLD. There were significant differences of score between patients who selected MLD and ELD in this group for wound size, anesthetic method, operative time, blood loss and length of stay (P< 0.05). In patients who had experience in discectomy group, 22 patients (76%) who underwent MLD still selected MLD if they could select surgical methods again for themselves, while 24 patients (96%) who underwent ELD still selected ELD if they could select again. The most important factor in patients who selected MLD was outcomes of treatment. The most important factor in patients who selected ELD was wound size. There were significant differences of scores between patients who selected MLD and ELD in this group for wound size, anesthetic method, operative time, complication, cost and length of stay (P< 0.05).

**Data Availability Statement:** All relevant data are within the manuscript and its Supporting Information files.

**Funding:** The author(s) received no specific funding for this work.

**Competing interests:** The authors have declared that no competing interests exist.

## Conclusions

About two thirds of the participants preferred ELD after reading the summary evidence information. The most important factor in MLD group was outcomes of treatment while the most important factor in ELD group was wound size.

## Introduction

The evolution of minimal invasive surgery has been widespread implemented in almost every surgery. With the development of new instrumentations, there are several advanced techniques which seem to improve the outcome of surgery. In spine surgery, there were evidences that suggested the outcome of minimal invasive spine surgery had significantly better outcomes than conventional surgery [1, 2]. Microscopic lumbar discectomy (MLD) and endoscopic lumbar discectomy (ELD) are 2 major minimal invasive spine surgeries for lumbar disc herniation. MLD was introduced by Caspar [3] and Yasagil [4]. The procedure consists of a small surgical wound, preserved paraspinal muscles and more targeted surgical exposure. It is considered a standard treatment due to the reduction of invasiveness and complications when compared with conventional open surgery. ELD was evolved from percutaneous chemonucleolysis which was firstly reported by Lyman Smith. Hijikata developed tubes which could pass through the posterolateral annulus of lumbar intervertebral disc under fluoroscopy. Percutaneous nucleotomy then was firstly done in 1975 [5]. Later, the first endoscopic views of a herniated nucleus pulposus were published by Kambin et al. [6] With the advanced technology of the camera and medical instruments, ELD is practical and gaining popularity. Several literatures have shown similar outcomes of ELD and MLD [7–9].

MLD is considered a current standard treatment with good results and more cost savings because no additional tools are needed. Although the treatment outcomes and complications were not significantly different, ELD can reduce tissue injuries and reduce hospitalization. The procedure also requires special equipment which increases the cost of surgery and skilled surgeons who have been trained to perform this kind of surgery.

The decision of selection between these two surgical procedures is usually unclear and depends on the doctor, patient, and the availability of the equipment in the hospital. To better understand the patients' perspective regarding their surgical preference, we surveyed the patients with spine problems by providing a brief and easy to understand evidence-based summary sheet and surveyed their opinions on why they selected the surgical method after reading the document.

## Materials and methods

### Participants

Cross sectional data survey was performed after approval for the study was obtained from Committee on Human Rights Related to Research Involving Human Subjects, Faculty of Medicine Ramathibodi Hospital, Mahidol University. Written informed consent was signed in all participants in order to answer the questionnaires. Patients who came for treatment at Spine Clinic, Department of Orthopaedics, Ramathibodi Hospital during August, 2018- July, 2019 were asked to participate in this study if they met the inclusion criteria. After providing informed consent, all respondents received the summary information sheet and survey

questionnaire. After the patients read the summary information sheet, they were asked to complete the anonymous questionnaire.

The inclusion criteria were the patients who had low back pain with or without radiculopathy caused by degenerative spine problems. We included the participants from our spine clinic, who never got spine surgery and those who underwent ELD or MLD at least 6 weeks ago. We excluded the patients who were diagnosed with spinal infection, spinal metastasis, inflammatory spine diseases and spine trauma.

## Summary information sheet and questionnaire

We reviewed the literature regarding the details of surgical techniques (benefits and drawbacks), anesthetic methods, wound size, operative time, blood loss, outcomes of surgery, complication rates, reoperation rates, estimated costs, and lengths of stay compared between ELD with MLD. The strength of the evidence was also considered according to the evidence in literature. (**See S1 Appendix**).

The summary information sheet was sent to 10 spine surgeons to evaluate the quality of the content and to evaluate bias of the content whether it favored or opposed any type of surgical methods. They were asked to classify the summary information sheet as "inappropriate", "appropriate after some adjustment", or "appropriate". The summary information sheet would be considered appropriate if the response was "appropriate" more than 70%, otherwise it was decided that the summary information sheet would be revised.

After being evaluated by orthopedic surgeons, the summary information sheet was sent to a group of patients to assess its understandability. They were asked to classify the summary information sheet as "do not understand", "partly understand", or "clearly understand". The summary information sheet would be considered understandable if the response was "clearly understand" more than 70%, otherwise it was decided that the summary information sheet would be revised.

We also created a questionnaire (**See S2 Appendix**) which consist of 3 parts. Part 1, the patients were asked background information such as sex, age, married status, education, occupation, income, and whether they had ever undergone spine surgery. If they had, what types of spine surgery which they underwent. Part 2, the patients were asked to complete 5 items survey regarding reasons to select types of surgery between MLD or ELD. They were asked about the important factors for determining the surgical decision making and assign a score from 1 to 10 based on how important those factors were (1, least important: 10, most important). Additionally, all participants were asked whether they had ever received information regarding surgical techniques of MLD or ELD. Also did the summary information sheet help the participants to select the types of surgery. Part 3, there were 4 items for the patients who had ever undergone either MLD or ELD. They were asked whether they had ever received information regarding to surgical techniques before surgery and did they have a chance to select the surgical methods. At this moment, if the patients had chance to choose the surgical methods again, do they choose the same operation. Also do they recommended the same surgical methods which they experienced to other people. Finally, the patients were asked for rating the confidence level of their answers.

## Statistical analysis

Descriptive statistics was used to present and summarize the categorical variables by frequency (n) and percentage (%). Inferential statistics were performed to examine whether selecting MLD or ELD related with their demographic and socioeconomic characteristics using the chi-squared test ($\chi$2) and T-test was used to analyze the income and Likert scaled questions, which

started from 1 to 10, that were analyzed to study which factors were significant for the patients to choose the operation. A p-value of less than 0.05 was considered to declare the statistically significance. Statistical analysis was performed using STATA software package, version 16.1 (Stata Corp, College Station, Texas, USA).

## Results

### Survey evaluation

All orthopedic surgeons who were asked to evaluate the summary information sheet cooperated and provided response. Nine out of ten (90%) classified the summary sheet as "appropriate", while the other orthopedic surgeon classified it as "appropriate after some adjustment". Also, eighty percent of the patients (12 out of 15) classified the summary sheet as "clearly understand". According to our pre-specified quality parameter, then the summary information sheet was considered appropriate for providing the knowledge to the patients.

### Participant characteristics

One hundred and sixty-one patients, aged from 17–89 years old, were included in the study. There were 60 (37.3%) males and 101 (62.7%) females who responded to the questionnaire. 63.4% was married, 29.2% was single and 6.8% was divorced. Majority of the participants about two-third were in the capital city and at least Bachelor degree graduation. Fifty-four participants had been operated before, by MLD (29 cases) or ELD (25 cases), and the remaining 107 participants had no experience in receiving spine surgery. The overall participants gave a confidence about the answer (1 = not strongly 10 = very strongly) at 8.32±1.62 (mean±SD) point scale.

### Participants' opinions regarding the selection of surgical methods

From a total of 161 patients, 107 patients (66.5%) selected endoscopic surgery and 54 patients (33.5%) selected microscopic surgery. We evaluated factors that could influence the selection between MLD and ELD. There was no statistically significant difference of demographic data between MLD selection patients and ELD selection patients including age, sex, married status, education, occupation, and income (p >0.05) (Table 1).

The 161 patients were asked to score how important each factor was for selecting the type of surgery, with scale from 1 (least important) to 10 (most important). The most important factor was outcomes of treatment 8.82±1.92 (mean±SD), followed by complications 8.09±2.33, wound size 7.88±2.39, length of stay 7.58±2.71, revision rate 7.57±2.50, blood loss 7.16±2.77, operative time 7.13±2.63, cost 6.96±2.58 and anesthetic method 6.62±2.58, respectively.

Fifty-three participants (32.9%) had ever received the information about surgical method, while most of them (67.1%) had not received this kind of information. They replied that the articles helped in decision making for selection of surgical methods as "Yes" in 59%, "Partially yes" in 36.6% and "No" in 4.3%.

### Opinion of participants who had no experience in lumbar discectomy

Among 107 patients who had no experience in lumbar discectomy, 76 patients (71%) selected ELD while 31 patients (29%) selected MLD. The most important factor in MLD group was outcomes of treatment with score 9.26±1.67 (mean±SD). The most important factor in ELD group factor was wound size with score 8.75±1.83 (mean±SD) (Fig 1).

There were significant differences of score between patients who selected MLD or ELD in this group for wound size, anesthetic method, operative time, blood loss and length of stay

**Table 1. Demographic data of participants in the study.**

| | Total Number (%) | MLD group | ELD group | P-value |
|---|---|---|---|---|
| **Age** | | | | |
| • < = 30 | 12 (7.5%) | 3 (5.6%) | 9 (8.4%) | 0.528 |
| • 31–40 | 21 (13%) | 8 (14.8%) | 13 (12.1%) | |
| • 41–50 | 26 (16.1%) | 6 (11.1%) | 20 (18.7%) | |
| • 51–60 | 44 (27.3%) | 18 (33.3%) | 26 (24.3%) | |
| • 61–70 | 42 (26.1%) | 12 (22.2%) | 30 (28.0%) | |
| • >70 | 16 (9.9%) | 7 (13.0%) | 9 (8.4%) | |
| **Sex** | | | | |
| • Male | 60 (37.3%) | 19 (35.2%) | 41 (38.3%) | 0.698 |
| • Female | 101 (62.7%) | 35 (64.8%) | 66 (61.7%) | |
| **Married status** | | | | |
| • Married | 102 (63.4%) | 38 (70.4%) | 64 (59.8%) | 0.658 |
| • Single | 47 (29.2%) | 13 (24.1%) | 34 (31.8%) | |
| • Divorced | 11 (6.8%) | 3 (5.6%) | 8 (7.5%) | |
| **Education** | | | | |
| • High school | 52 (32.3%) | 20 (37.0%) | 32 (29.9%) | 0.575 |
| • Bachelor degree | 79 (49.1) | 27 (50.0%) | 52 (48.6%) | |
| • Master degree | 27 (16.8) | 6 (11.1%) | 21 (19.6%) | |
| • Higher than Master degree | 3 (1.9) | 1 (1.9%) | 2 (1.9%) | |
| **Occupation** | | | | |
| • Student | 3 (1.9) | 1 (1.9%) | 2 (1.9%) | 0.569 |
| • Merchandiser | 39 (24.2) | 17 (31.5) | 22 (20.6%) | |
| • Employee | 12 (7.5) | 4 (7.4%) | 8 (7.5%) | |
| • Government officer | 60 (37.3) | 16 (29.6%) | 44 (41.1%) | |
| • Other | 47 (29.2) | 16 (29.6%) | 31 (29.0%) | |
| | **Mean ± SD** | **Microscope selection group** | **Endoscope selection group** | **P-value** |
| **Income/month (USD)** | 880.01 ±567.71 | 814.51 ±587.75 | 907.66 ±560.07 | 0.398 |

MLD, Microscopic Lumbar Discectomy; ELD, Endoscopic Lumbar Discectomy

(p < 0.05). All of these factors' scores were higher in ELD selected group than in MLD selected group (Fig 1).

## Opinion of participants who had experience in lumbar discectomy

In 54 patients who had experience in lumbar discectomy, 29 patients underwent MLD and 25 patients underwent ELD. When the patients were asked if they would select the surgical methods again, those patients who had been operated with MLD, 22 patients (76%) still selected the MLD and 7 patients (24%) selected ELD. While patients who had been operated with ELD, 24 patients (96%) still selected ELD and 1 patient (4%) selected MLD (Table 2). The most important factor in MLD group was outcomes of treatment with score 9.34±1.11 (mean±SD). The most important in ELD group factor was wound size with score 9.28±1.27 (mean±SD) (Fig 1).

There were significant differences of scores between patients who selected MLD or ELD in this group for wound size, anesthetic method, operative time, complication, cost and length of stay (p < 0.05). All of these factors' scores were higher in ELD selected group than in MLD selected group except for the complication, in which patients in MLD group scored higher than patients in ELD group (Fig 2).

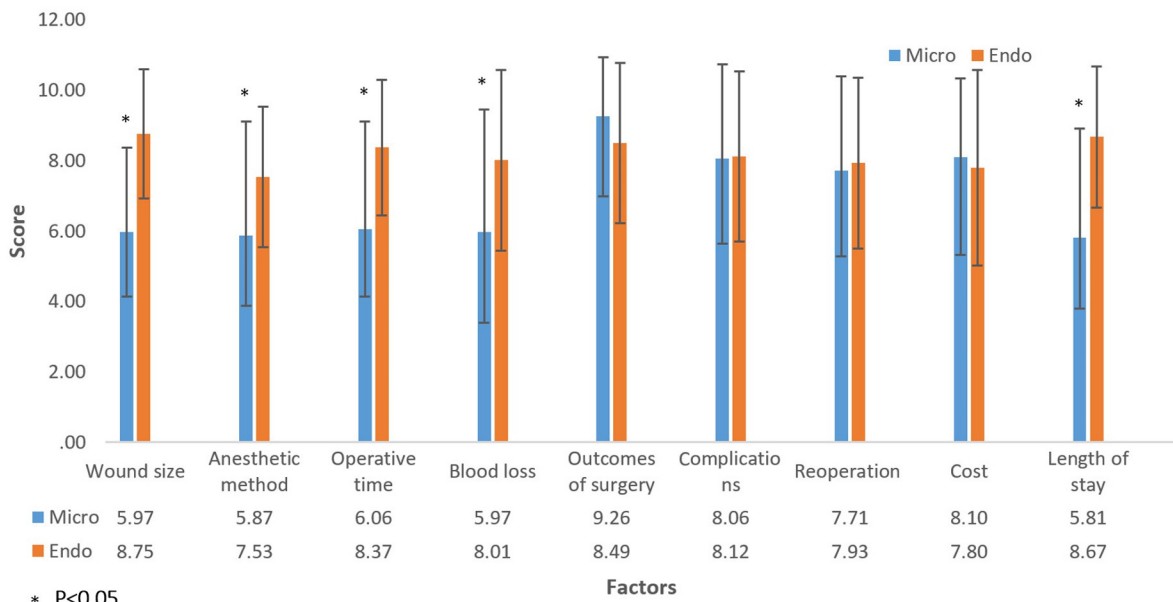

**Fig 1. Opinions of the participants who had no experience in lumbar discectomy for how important of these factors to select the microscopic lumbar discectomy or endoscopic lumbar discectomy [Scale from 1 (least important) to 10 (most important)].**

In patients who had been operated (54 patients), 28 (52.2%) stated that they received the information regarding to ELD or MLD before the surgery, and 33 participants (61.1%) had a chance to select the surgical treatment methods. If they had a chance to select surgical treatment methods again, 46 participants (85.2%) would select the same operation. Additionally, 51 participants (94.4%) would recommend the same operation to other people who need operative treatment.

## Discussion

There are several studies comparing MLD and ELD for lumbar disc herniation [5, 7, 8]. According to the randomized controlled study of Ruetten et al, both treatments showed comparable significantly improved clinical outcome when compared by Visual Analog Scale (VAS) pain scores, Oswestry Disability Index (ODI) scores and North American Spine Society Instrument scores. ELD was performed with shorter operative time, lesser intraoperative blood loss, and fewer complication rates when compared with MLD, while the reoperation rate was not different [10]. In contrast, Chen et al. reported over the 2-year follow-up period ELD did not show superior clinical outcomes and did not seem to be a safer procedure for patients with lumbar disc herniation compared with microendoscopic discectomy. ELD had

**Table 2. Operative selection of the participants which has been classified by previous treatment received.**

| If you have to undergo lumbar discectomy, what kind of surgical treatment do you want to choose? | | Treatment received | | | Total |
|---|---|---|---|---|---|
| | | No previous spine surgery | MLD | ELD | |
| | **MLD** | 31 | 22 | 1 | 54 |
| | **ELD** | 76 | 7 | 24 | 107 |
| | **Total** | 107 | 29 | 25 | 161 |

MLD, Microscopic Lumbar Discectomy; ELD, Endoscopic Lumbar Discectomy

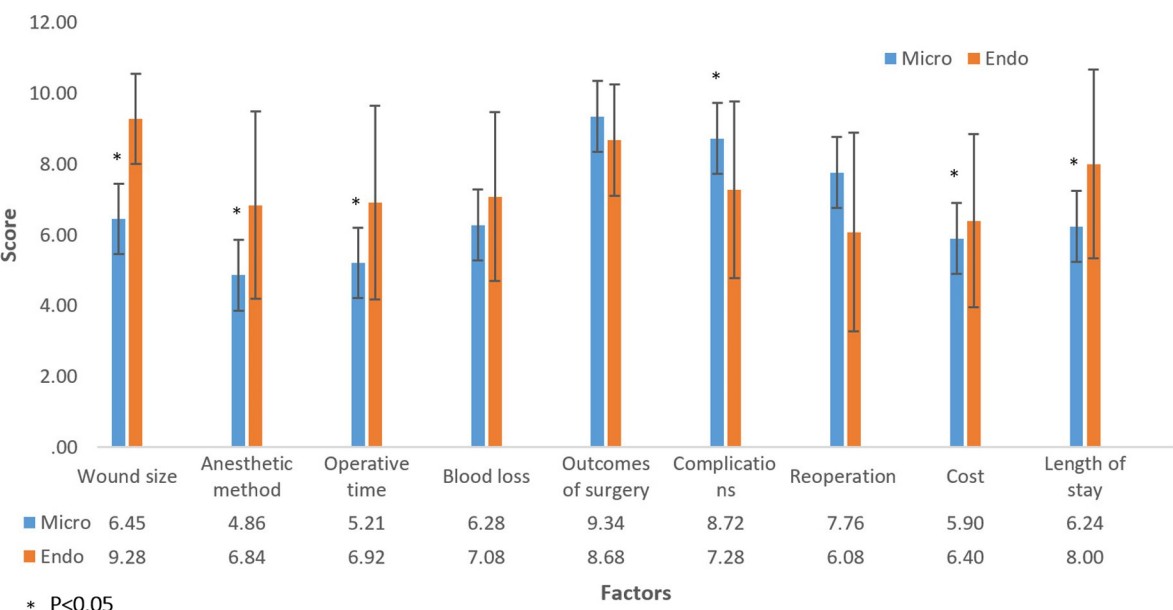

**Fig 2. Opinions of the participants who had experience in lumbar discectomy for how important of these factors to select the microscopic lumbar discectomy or endoscopic lumbar discectomy [Scale from 1 (least important) to 10 (most important)].**

inferior results for median disc herniation and had superior outcomes for far-lateral disc herniation. The reasons were narrow neural foramen and small working cannula of endoscope had an effect on difficultly to completely remove large central disc herniation [11].

Many systematic review and meta-analysis studies have been conducted. Ruan et al. found that both methods showed no significant difference in pain reduction score (P = 0.95). The incidence of complications (P = 0.07) and reoperation rate (P = 0.09) were not significantly different between the two methods. ELD when compared with MLD, required less duration for surgery (P = 0.04) and shorter hospital stay (P = 0.0006) [12]. The similar results were confirmed by the study of Qin et al. [8]. They found no significant differences existed in VAS and ODI scores between ELD and open MLD. Both procedures were also similar in terms of operative time, complications, and incidence of recurrence and reoperation, but ELD showed shorter hospital stay and time of return to work [8]. Based on previous evidences, we remarked the similar clinical outcomes between these 2 methods such as pain and functional score improvement. However, there was a trend that ELD might give some improvement in some perioperative outcomes such as blood loss, operative time and length of hospital stay.

Variety of surgical options for lumbar disc herniation have been proposed such as endoscopic, laser disc decompression, microscopic and microendoscopic methods, and the selection depends on the pathology, surgeon's preference and availability of equipment [13]. Patient consideration is playing a major part in decision making, particularly in the controversial point of care, although this aspect is not well studied and is ignored in the treatment consideration of lumbar disc herniation.

From the overall results of this study, 32.9% of the participants had ever received the information of different discectomy methods. In the group of previous surgery more percentage 52.2% had heard about this information, but only 61.1% had a chance to select the treatment. Nevertheless, 85.2% would select the same operation and 94.4% would recommend the same operation to other people who need operative treatment. This revealed that both MLD and ELD should provide acceptable results and satisfaction of the surgery for the patients.

About two thirds of the participants preferred endoscopic spine surgery after reading the provided information. We analyzed the factors which influenced the selection of discectomy methods. There were no statistically significant differences of basic demographic data between MLD group and ELD group, which revealed these factors did not influence the selection of surgical methods. Surgeon preference of MLD or ELD may not be decided by the demographic data of the patient such as lower or higher socioeconomic or education.

When the patients who had experience in lumbar discectomy were asked to select the operation again, 76% of the patients who have had MLD still selected MLD again. These group of patients scored the outcomes of treatment was the most important factors for selecting the operation. Meanwhile, 96% of the patients who have had ELD will select ELD again. Wound size is the most important factor in this group of patients. From the summary of the information sheet, most of the advantages seemed like ELD may be superior to MLD in many aspects especially less invasive, although the outcomes of treatment were not significantly different. This may cause most of the patients to select ELD for their operation. However, in the MLD experienced patients who went through the operation, most of them selected MLD again, and did not give priority about invasiveness such as wound size, anesthetic method, operative time, length of stay. Instead, they concerned about the complications. These results showed that the opinions and concerns of the patients were different in perioperative phases of treatment.

In 2017, Gadjradj et al. reported the international survey of surgeons from 89 countries according to current practice for lumbar herniation. The results showed surgeon's expectation for the best surgical methods should provide the highest effectiveness and the lowest risk for complications. Our study showed patients' perspective for surgical methods should provide the best outcomes and less invasive. These results showed both similar and different opinions between surgeon's perspective and patient's perspective [14].

Shared decision-making has received increased attention as a process to reinforce patient-centered care [15], and has been used in various diseases such as oncology and neonatology medicine [16, 17]. It should be considered when controversial treatment divided clinicians' opinion and the different treatments impact on the patient outcomes. It might provide the health decision for best available evidence and patient's preference [18]. Our results showed only about one-third of participants and half of the patients who received previous surgery had the information of different surgical technique preoperatively. If possible, the concept of shared decision-making should be advocated more in the institution where the treatment options were available.

There are some limitations in our study. Firstly, the participants were included from a single institution which could not give generalizability, and the previous surgery patients might have a recall bias, short term follow up, also the results of surgery may influence questionnaire response. Secondly, we provided summary medical evidences and understandable information of treatment options, although some surveyed participants might not fully understand the summation which possibly impacted on the answers. Thirdly, we did not analyze that patient's preference really affected the outcomes of surgery. Even the patient selected endoscopic or microscopic surgery, it might not impact on clinical or functional outcome. It still cannot be concluded that patient's preference is an important factor affecting the good surgical outcome in lumbar disc herniation.

## Conclusions

About two thirds of the participants preferred endoscopic spine surgery after reading the summary evidence information. Basic demographic data does not influence the surgical method selection. The most important factor in MLD group was outcomes of treatment while the most

important factor in ELD group was wound size. Most of the patients who had experience in lumbar discectomy will select the same operation again and will recommend the same operation to other patients. This study emphasized the patients' perspective after they received evidence-based information for selecting surgical methods.

## Supporting information

**S1 Appendix. Surgical treatment options for lumbar disc herniation.**
(DOCX)

**S2 Appendix. Survey questionnaire.**
(DOCX)

## Author Contributions

**Conceptualization:** Gun Keorochana, Chaiwat Kraiwattanapong.

**Data curation:** Gun Keorochana, Chaiwat Kraiwattanapong.

**Formal analysis:** Umaporn Udomsubpayakul, Tulyapruek Tawonsawatruk.

**Investigation:** Gun Keorochana, Chaiwat Kraiwattanapong, Thamrong Lertudomphonwanit, Pittavat Leelapattana, Pongsthorn Chanplakorn, Nutthee Wannaratsiri.

**Methodology:** Gun Keorochana.

**Project administration:** Chaiwat Kraiwattanapong.

**Writing – original draft:** Gun Keorochana.

**Writing – review & editing:** Chaiwat Kraiwattanapong, Thamrong Lertudomphonwanit.

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
