## [Decision Letter · Decision Letter 0]

23 Jan 2023

PONE-D-22-08950Survey research of patient’s preference on choosing microscopic or endoscopic spine surgery for lumbar discectomy.PLOS ONE

Dear Dr. Kraiwattanapong,

Thank you for submitting your manuscript to PLOS ONE. After careful consideration, we feel that it has merit but does not fully meet PLOS ONE’s publication criteria as it currently stands. Therefore, we invite you to submit a revised version of the manuscript that addresses the points raised during the review process.

We look forward to receiving your revised manuscript.

Kind regards,

Thamer Hamdan, PhD

Academic Editor

PLOS ONE

Journal Requirements:

a) Did participants provide their written or verbal informed consent to participate in this study?

Reviewers' comments:

Reviewer's Responses to Questions

**Comments to the Author**

1. Is the manuscript technically sound, and do the data support the conclusions?

Reviewer #1: Yes

Reviewer #2: Partly

2. Has the statistical analysis been performed appropriately and rigorously? 

Reviewer #1: Yes

Reviewer #2: Yes

3. Have the authors made all data underlying the findings in their manuscript fully available?

Reviewer #1: Yes

Reviewer #2: Yes

4. Is the manuscript presented in an intelligible fashion and written in standard English?

Reviewer #1: Yes

Reviewer #2: No

5. Review Comments to the Author

Reviewer #1: Thank you for giving me the chance to review the manuscript entitled “Survey research of patient’s preference on choosing microscopic or endoscopic spine surgery for lumbar discectomy”. This is a very interesting and useful study. The experiment was well designed. And the manuscript was well written. It could promote doctors getting a clearer idea of patients’ preferences for surgical approaches.

1. As a spine surgeon, I think some patients are not suitable for ELD, whether such patients excluded from this study?

2. I think the author could include patients who underwent surgery for one year or more years in future studies. It can help to understand how the long-term outcomes of the two surgical methods influence patients' preferences.

3. The conclusion is too long, which should be shortened.

Reviewer #2: Dear respected authors. Congratulation for your work. Few remarks for the purpose of perfection I will mention:

*Line 39: most patients prefer ELD (patient preference]

Line 62: ELD had same outcome with less complications than MLD [scientific preference].

Therefore, you should add in the conclusion that ELD is preferred scientifically and by patient’s preference and no need to ask for patient’s opinion in future, as ELD is preferred whenever it is available and possible.

Line 22, 65: There is no clear evidence for how to select the procedures. Why? (You mention later in 61- 62, line 208, 221 [Although the treatment outcomes and complications were not significantly different, ELD can reduce tissue injuries and reduce hospitalization].

*Line 138: what is the value of including the marital status, residency and academic degree in the questionnaire of this study?

*Where are the patient information sheath? Like what is differences about wound size, time of surgery … between the two procedures in the information sheath. These information of course affect patient decision.

*Page 29, 30, 31: links: readers should read all informations of the subject in one file, not to follow a links. Please try to merge informations of the appendicular files with the main manuscript. It is difficult to publish this form of paper.

6. PLOS authors have the option to publish the peer review history of their article (what does this mean?). If published, this will include your full peer review and any attached files.

Reviewer #1: No

Reviewer #2: **Yes: **Raed J. Chasib

---

## [Author Response · Author response to Decision Letter 0]

9 Feb 2023

Response to reviewers as follow: 

Reviewer #1: 

Comment 1: As a spine surgeon, I think some patients are not suitable for ELD, whether such patients excluded from this study?

Reply: In our institute we definitely have exclusion criteria for endoscopic discectomy such as cauda equina syndrome, spondylolisthesis, segmental instability and large calcified disc herniation. However, our study was cross sectional questionnaire study which included non-surgical and previous surgical cases thus we did not explain in details for indication or contraindication of microscopic or endoscopic spine surgery.

Comment 2: I think the author could include patients who underwent surgery for one year or more years in future studies. It can help to understand how the long-term outcomes of the two surgical methods influence patients' preferences.

Reply: Thank you very much for your comment. We agreed and added this in our limitation in Discussion. Further study for long term outcomes of the two surgical methods are very interesting. 

Comment 3: The conclusion is too long, which should be shortened.

Reply: We revised our conclusion which made it more concise as yours suggestion.

………………………………………………………………………………………………………….

Reviewer #2: 

Comment1:

Line 39: most patients prefer ELD (patient preference]

Line 62: ELD had same outcome with less complications than MLD [scientific preference].

Therefore, you should add in the conclusion that ELD is preferred scientifically and by patient’s preference and no need to ask for patient’s opinion in future, as ELD is preferred whenever it is available and possible.

Comment2:

Line 22, 65: There is no clear evidence for how to select the procedures. Why? (You mention later in 61- 62, line 208, 221 [Although the treatment outcomes and complications were not significantly different, ELD can reduce tissue injuries and reduce hospitalization].

Reply: Both ELD and MLD are minimal invasive spine surgery. From the literatures, the primary outcomes such as pain, function score, reoperation and complication are not statistically significant difference. The ELD may be more advantage in term of length of stay and blood loss as mention in Discussion. However, the different between them is not enough to affect the outcomes of the treatment such as the different in blood loss is about 45 cc [Reutten S et al Spine (Phila Pa 1976) 2008;33:931-9.] and no need to transfusion. Then we could not clearly advice the patients that which operation is better. That why we asked the patients opinion, especially, the patients who had experience of surgery compared with patients who had no experience of surgery and received information only from the documents. 

Comment 3: Line 138: what is the value of including the marital status, residency and academic degree in the questionnaire of this study?

Reply: We hypothesize that some demographic data would affect the patient’s decision such as socioeconomic, family status or educational level. There are many information and methods of spine surgery can be found online which could affect the preference of patients. However, our results showed that the demographic data did not statistically influence the patient’s selection. 

Comment 4: Where are the patient information sheath? Like what is differences about wound size, time of surgery … between the two procedures in the information sheath. These information of course affect patient decision.

Reply: The comparison information sheet between MLD and ELD was in Appendix1.

Comment 5: Page 29, 30, 31: links: readers should read all informations of the subject in one file, not to follow a links. Please try to merge informations of the appendicular files with the main manuscript. It is difficult to publish this form of paper.

Reply: There are many details in the information sheet and the questionnaire which could make the main manuscript too long. We would like to create links that can access the appendix 1 and the appendix 2 when click on the words.

---

## [Decision Letter · Decision Letter 1]

21 Mar 2023

Survey research of patient’s preference on choosing microscopic or endoscopic spine surgery for lumbar discectomy.

PONE-D-22-08950R1

Dear Dr. Kraiwattanapong,

We’re pleased to inform you that your manuscript has been judged scientifically suitable for publication and will be formally accepted for publication once it meets all outstanding technical requirements.

Kind regards,

Thamer Hamdan, PhD

Academic Editor

PLOS ONE

Additional Editor Comments (optional):

Reviewers' comments:

Reviewer's Responses to Questions

**Comments to the Author**

1. If the authors have adequately addressed your comments raised in a previous round of review and you feel that this manuscript is now acceptable for publication, you may indicate that here to bypass the “Comments to the Author” section, enter your conflict of interest statement in the “Confidential to Editor” section, and submit your "Accept" recommendation.

Reviewer #2: All comments have been addressed

2. Is the manuscript technically sound, and do the data support the conclusions?

Reviewer #2: Yes

3. Has the statistical analysis been performed appropriately and rigorously? 

Reviewer #2: Yes

4. Have the authors made all data underlying the findings in their manuscript fully available?

Reviewer #2: Yes

5. Is the manuscript presented in an intelligible fashion and written in standard English?

Reviewer #2: Yes

6. Review Comments to the Author

Reviewer #2: (No Response)

7. PLOS authors have the option to publish the peer review history of their article (what does this mean?). If published, this will include your full peer review and any attached files.

Reviewer #2: No

---

## [Editor Report · Acceptance letter]

27 Mar 2023

PONE-D-22-08950R1 

Survey research of patient’s preference on choosing microscopic or endoscopic spine surgery for lumbar discectomy. 

Dear Dr. Kraiwattanapong:

I'm pleased to inform you that your manuscript has been deemed suitable for publication in PLOS ONE. Congratulations! Your manuscript is now with our production department. 

Kind regards, 

on behalf of

Professor Thamer Hamdan 

Academic Editor

PLOS ONE